# DYNAMICS OF INSTRUCTION TUNING: EACH ABILITY OF LARGE LANGUAGE MODELS HAS ITS OWN GROWTH PACE

## ABSTRACT

Instruction tuning is a burgeoning method to elicit the general intelligence of Large Language Models (LLMs). However, the creation of instruction data is still largely heuristic, leading to significant variation in quality and distribution across existing datasets. Experimental conclusions drawn from these datasets are also inconsistent, with some studies emphasizing the importance of scaling instruction numbers, while others argue that a limited number of samples suffice. To better understand data construction guidelines, we deepen our focus from the overall model performance to the growth of each underlying ability, such as creative writing, code generation, and logical reasoning. We systematically investigate the effects of data volume, parameter size, and data construction methods on the development of various abilities, using hundreds of model checkpoints (7b to 33b) fully instruction-tuned on a new collection of over 40k human-curated instruction data. This proposed dataset is stringently quality-controlled and categorized into ten distinct LLM abilities. Our study reveals three primary findings: (i) Despite data volume and parameter scale directly impacting models' overall performance, some abilities are more responsive to their increases and can be effectively trained using limited data, while some are highly resistant to these changes. (ii) Human-curated data strongly outperforms synthetic data from GPT-4 in efficiency and can constantly enhance model performance with volume increases, but is unachievable with synthetic data. (iii) Instruction data brings powerful cross-ability generalization, with evaluation results on out-of-domain data mirroring the first two observations. Furthermore, we demonstrate how these findings can guide more efficient data constructions, leading to practical performance improvements on public benchmarks.

## 1 INTRODUCTION

Large Language Models (LLMs) have shown impressive capabilities across diverse tasks (Brown et al., 2020; Touvron et al., 2023; Chowdhery et al., 2022; Almazrouei et al., 2023; Wang et al., 2022a; Wei et al., 2022b; Zhao et al., 2021; Wei et al., 2023; Ivison et al., 2022; Zhang et al., 2023b; Radford et al., 2019), demonstrating their potential for artificial general intelligence (Bubeck et al., 2023). A key contributor to this success is instruction tuning, a process involving supervised fine-tuning of LLMs on instruction-output pairs (Ouyang et al., 2022; Taori et al., 2023; Chiang et al., 2023; Iyer et al., 2022; Zhou et al., 2023). Despite the acknowledged significance of instruction tuning, there remains a lack of consensus on what type of instruction data is more beneficial. For instance, while some studies (Wei et al., 2022a; Sanh et al., 2021) argue that scaling data volume is crucial for success, other results (Zhou et al., 2023) suggest a limited number of instructions is sufficient, with models tuned on mere 1k instances outperforming those tuned on other datasets that are 10x-100x larger.

Intuitively, the instruction data develops a wide range of abilities for LLMs, such as creative writing, code generation, and logical reasoning, each with a different level of intelligence. Thus, we hypothesize that the inconsistency observed across various studies stems from how different abilities, shaped by instruction tuning, respond unevenly to alterations in factors like data volume. To validate our

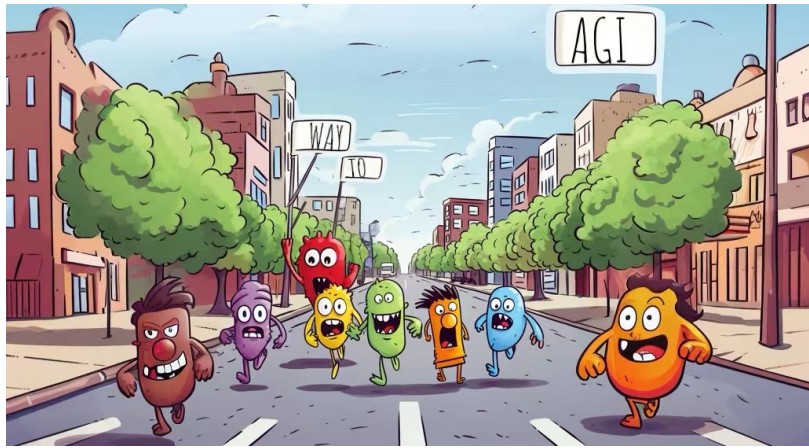

Figure 1: Each ability of LLMs has its own growth pace during instruction tuning.

hypothesis, we systematically investigate the growth of each underlying ability against data volume, parameter size, and data construction methods.

To this end, we propose a new dataset encompassing over 40,000 human-curated instruction instances, covering ten distinct LLM abilities. Each data instance is rigorously revised by annotators to ensure high-quality text and is categorized into an individual ability, allowing us to analyze each ability of instruction-tuned models against data volume.

Using our proposed dataset, we instruction-tune hundreds of model checkpoints, ranging from 7b to 33b parameters, and our results reveal three primary findings on instruction tuning:

1. Data quantity or parameter size significantly influences overall performance, but each ability develops at different paces during instruction tuning. Abilities such as creative writing are more responsive to these factors and can be well-trained with a small amount of data. In contrast, abilities like ethics show resistance to these changes, suggesting that alternative approaches beyond supervised fine-tuning may be necessary for their development.

2. Regarding data construction methods, synthetic data from GPT-4 falls short in performance for instruction tuning. Compared to our human-curated data, synthetic data is less efficient and fails to consistently enhance model performance by increasing its volume.

3. Instruction data promotes powerful cross-domain generalization, benefiting abilities beyond those included in our dataset. Evaluation results on out-of-domain (OOD) data support the observations made in the first two findings, with different OOD abilities showing distinct growth rates and human-curated data proving more helpful than synthetic data.

Guided by these findings, we adjust the quantity of different ability data and the mixing proportion of synthetic data, achieving practical performance gains on two public benchmarks, CMMLU (Li et al., 2023) and AGIEval (Zhong et al., 2023). We open-source our codebase, dataset, and model checkpoints for reproducibility and future research[1].

## 2  RELATED WORK

Instruction datasets are crucial for the efficacy of instruction-tuned large language models, and their construction methods can be broadly categorized into three types: **Task-formatted datasets** (Sanh et al., 2021; Muennighoff et al., 2022; Wei et al., 2022a; Chung et al., 2022; Mishra et al., 2021; Wang et al., 2022c) incorporate instances from various NLP tasks, including text summarization, natural language inference, sentiment classification, and many other supervision tasks. A sign of these datasets is the inclusion of human-written templates to format each task, such as *"Please summarize the following text:"* These descriptions differentiate task instances so that models can be trained in a multi-task manner. To better scale up the data volume, a crowd-sourcing platform called PromptSourceBach et al. (2022) has also been developed to facilitate the sharing and reviewing

---

[1]The link to be added.

Table 1: The data sources, data size, and annotation procedures for each ability category.

| Ability | Data Source | Data Size | Annotation Procedure | | |
|---|---|---|---|---|---|
| | | | Standardization | Human Filtering | Human Revision |
| STEM - Biology | COIG - Exam (Zhang et al., 2023a) | 1,242 | ✓ | ✓ | ✓ |
| Humanity - History | COIG - Exam (Zhang et al., 2023a) | 2,093 | ✓ | ✓ | ✓ |
| Code Generation | Leetcode.cn | 5,168 | ✓ | ✓ | ✗ |
| Creative Writing | User Queries from In-House Data | 1,200 | ✓ | ✓ | ✓ |
| Chinese | COIG - Exam (Zhang et al., 2023a) | 1,650 | ✓ | ✓ | ✓ |
| Dialogue Understanding | C3-D (Sun et al., 2020) | 5,085 | ✓ | ✓ | ✗ |
| Role-play Chat | BELLE (Ji et al., 2023) | 1,200 | ✓ | ✓ | ✓ |
| Logical Reasoning | LogiQA2.0 (Liu et al., 2023) | 12,951 | ✓ | ✓ | ✗ |
| COT for Grad-Math | PRM800K (Lightman et al., 2023) | 9,201 | ✓ | ✓ | ✗ |
| Ethics | COIG - Human Value (Zhang et al., 2023a) | 1,200 | ✓ | ✓ | ✓ |

of task descriptions. Even though this method can easily enlarge data size in early studies, we focus on methods other than it due to its potential mismatches with actual human requests (Ouyang et al., 2022; Zhao et al., 2023). **Human-curated datasets** (Ouyang et al., 2022; Zhou et al., 2023; Conover et al., 2023; Köpf et al., 2023) address the issue above using genuine user queries, website Q&As, examination questions, and other sources of real-life tasks to construct instructions. Human labelers are assigned to provide output by revising the given answer or directly answering these instructions. Proprietary models like ChatGPT (OpenAI, 2022) and GPT-4 (OpenAI, 2023) employ these procedures to generate training data. Unfortunately, most of these datasets are kept in-house due to the costly and time-consuming nature of the collection process. **Synthetic datasets** (Honovich et al., 2022; Xu et al., 2023a;b) reduce the need for manual collection and annotation of instructions through semi-automated approaches. One approach is collecting user chats with well-performed proprietary models, as in ShareGPT[2]. Another representative approach is Self-Instruct (Wang et al., 2022b), which only requires approximately 100 seed tasks to initiate. It leverages well-performed LLMs to update the task pool as well as generate instruction-output pairs. These explorations effectively synthesize large-scale data at a low budget and encourage open reproductions such as Alpaca (Taori et al., 2023) and Vicuna Chiang et al. (2023).

The influence of dataset construction methods on instruction tuning has been a subject of debate. Existing literature presents a range of findings: Wei et al. (2022a) and Chung et al. (2022) advocate that larger training datasets significantly enhance model performance. In contrast, Zhou et al. (2023) argues that a well-trained model can be achieved with a limited number of human-curated data. Sanh et al. (2021) and Wei et al. (2022a) propose that instruction-tuned models generalize effectively to unseen tasks, whereas Gudibande et al. (2023) suggests that these models only excel in tasks heavily supported in the training dataset. Several studies have also explored various data mix strategies, including balancing data distribution (Longpre et al., 2023), examples-proportional mixing with maximum caps (Raffel et al., 2019; Wei et al., 2022a), and combining multiple datasets (Iyer et al., 2022; Wang et al., 2023).

The inconsistencies observed in these studies have inspired us to investigate the growth pace of various underlying abilities during instruction tuning. In Section 4, we identify significant disparities in the impact of data on different abilities: some abilities show high responsiveness to data increases and can be effectively trained with limited data. Conversely, some abilities are data-hungry, requiring much more data for effective training. This new perspective provides a potential explanation for the differing conclusions drawn from existing research.

## 3  A New Collection of Instruction Data

To systematically investigate the roles of data quantity, parameter size, and data construction methods in shaping a range of model abilities, it is necessary to rule out the influence of data quality and establish a controllable data distribution. Current instruction datasets are insufficient for these research needs. Hence, we introduce a new human-curated Chinese dataset, comprising more than 40,000 instruction-output pairs, each subject to stringent quality control. Moreover, each instance is explicitly categorized into one of ten ability types, enabling us to easily balance or mix them to meet specific experimental proportions. This section will introduce the annotation process for our instruction data.

---

[2]https://sharegpt.com/

Following the previous studies in Section 2, our human-curated data originates from real-life scenarios such as academic examinations, online platforms, and user queries. We organize this data into ten representative ability categories: (1) STEM subject - Biology, (2) Humanity subject - History, (3) Code Generation, (4) Creative Writing, (5) Language proficiency - Chinese, (6) Dialogue Understanding, (7) Role-play Chat, (8) Logical Reasoning, (9) Chain of Thought, and (10) Ethics.

Data from diverse sources significantly differ in format, including raw web pages, exam papers, user inputs, and data cleaned by other researchers to different extents. To ensure consistent quality across all data instances, we employ a three-stage annotation process:

1. **Standardization**: In this stage, we standardize the raw data of different formats into uniform instruction-output pairs through programs. We customize different rules for each data category to extract valid text and eliminate duplicates. Notably, the data for "Chain of Thought" is sourced from PRM800K (Lightman et al., 2023), the only non-Chinese source, and is translated using the ChatGPT (OpenAI, 2022) API before human review.

2. **Human Filtering**: At this stage, each data instance is reviewed by two independent annotators. They are required to (i) Check the correctness of the text. (ii) Control the diversity of instructions, such as filtering out high-frequency personas in Role-play Chat. (iii) Avoid potential ethical issues in the output, such as biased opinions in Creative Writing. A data instance is marked as "pass" only if both annotators approve it, otherwise marked as "fail." In statistics, the pass rates for different ability categories range from 22.8% to 98.3%.

3. **Human Revision**: For categories with a small base in quantity or low pass rate, we conduct human revision to ensure sufficient numbers for experiments. Each question is revised or directly answered by an annotator, and then the answer undergoes the same process as in stage 2, with two additional reviewers determining its validity.

All the hired annotators are native Chinese speakers and hold a bachelor's degree or higher. The entire annotation process requires over 1,000 labor hours. To meet the experimental requirements in Sec 4, 1000 training data, 100 validation data, and 100 test data for each ability are annotated in the first round. We then expand the training set to 40k to compare different construction strategies in Sec 5. The data sources, data volume, and cleaning procedure for different ability data are outlined in Table 1. In the Appendix A.1, we also presented examples of each ability category.

## 4 EXPERIMENTS

Employing the human-curated dataset proposed in Section 3, we systematically study the abilities' development in response to alterations in data volume, parameter size, and construction methods. We conduct experiments under both in-domain and out-of-domain conditions. This section outlines the process of model training, evaluation, and results analysis.

### 4.1 EXPEIMENT SETUP

For quantity-based experiments, we uniformly sample data $d_i$ of size $n$ from each ability $a_i$ within the ten categories $A = \{a_1, a_2, ..., a_{10}\}$ in our training set. The samples, combined as $D = \bigcup_{i=1}^{10}\{d_i\}$, are utilized for each model training. We increment the sample size from $n = 1$ logarithmically (base 4) to $n = 1000$ (totaling 10k instances). Regarding parameter sizes, we train models across a full range of 7b, 13b, and 33b scales. To compare different data construction methods, we also examine models trained on synthetic data from GPT-4 (Peng et al., 2023), extending the data volume to 41k instances at 7b and 13b scales. Each training session spans at least 15 epochs, with the corresponding checkpoint saved for evaluation after each epoch. Our study, incorporating all these factor changes, consists of nearly 500 model checkpoints to draw systematic experimental conclusions.

Our hyperparameter choices are generally in line with Zhou et al. (2023), using AdamW optimizer with $\beta_1 = 0.9$, $\beta_2 = 0.95$ and $weight\_decay = 0.1$. The peak learning rate is set to $1e$-5 and linearly decays to $1e$-6 by the end of training. The batch size is 64, with inputs exceeding 2048 tokens trimmed. Referring to the protocol in Song et al. (2023), we also incorporate two speaker tokens, $< user >$ and $< assistant >$, to segment the utterances from instruction and output for training and inference. The foundation model we use is Chinese-LLaMA (Cui et al., 2023), a series of LLaMA (Touvron et al., 2023) models continuously pre-trained on a 120G Chinese corpus.

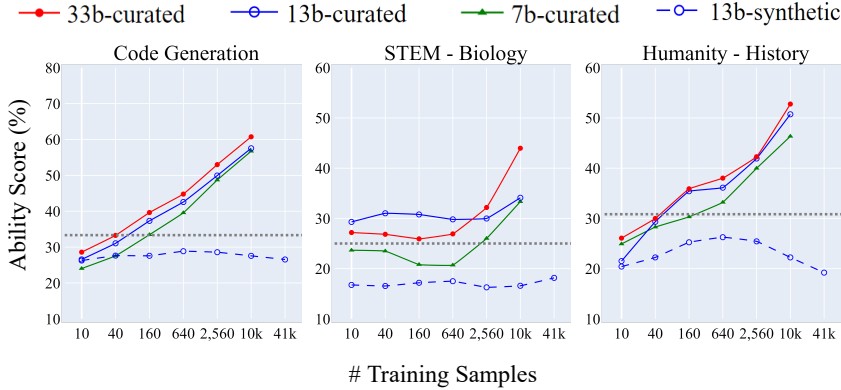

Figure 2: Abilities that are responsive to the data quantity and parameter scale on human-curated data, also comparing the data efficiency of different construction methods with synthetic data.

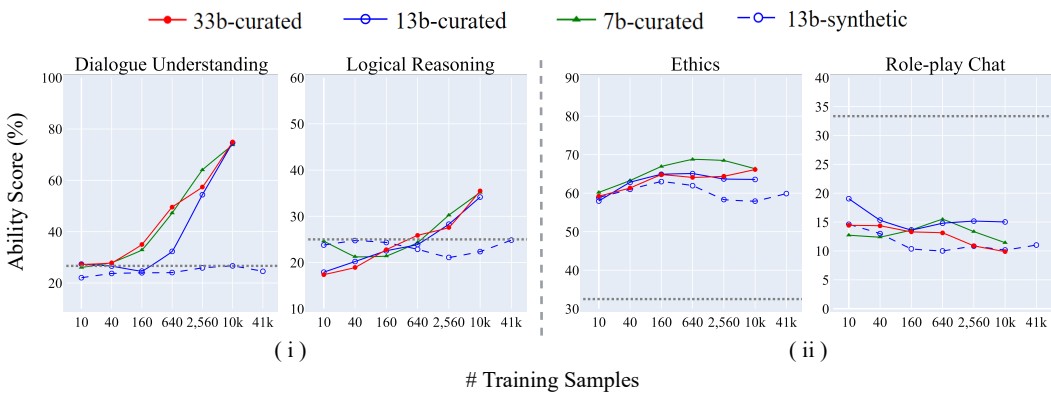

Figure 3: Abilities that are resistant to (i) parameter scale and (ii) both parameter scale and data volume on human-curated training data, also comparing the data efficiency of different construction methods with synthetic data.

## 4.2 EVALUATION

Selecting the optimal checkpoint for instruction-tuning is non-trivial. Prior studies (Ouyang et al., 2022; Zhou et al., 2023) note that training for more epochs can enhance the model's capabilities despite the risk of overfitting, and usually employ humans for evaluation. In contrast, automated evaluation is a more scalable solution but has long-lasting concerns about reliability. Despite recent improvements in LLM-based evaluators over traditional metrics (Papineni et al., 2002; Lin, 2004; Banerjee & Lavie, 2005), issues such as false reasoning processes and flawed instruction adherence (Luo et al., 2023; Shen et al., 2023; Chiang & Lee, 2023) persist. Therefore, to efficiently and accurately scale the evaluation across hundreds of checkpoints, we employ a semi-automated approach to reduce the burden on human annotators.

There are two types of questions in our dataset that correspond to distinct evaluation approaches:

- Exact-match questions, such as multiple-choice, true/false, and fill-in-the-blank questions. Each question has only one exclusive gold answer. Similar to other public benchmarks (Hendrycks et al., 2020; Li et al., 2023; Huang et al., 2023; Zhong et al., 2023), we automatically compute the accuracy by comparing generated answers to the ground truth.

- Open-ended questions are common in creative writing, role-play chat, and code generation abilities. These questions do not have standard answers. We thus propose a semi-automated "comparison with distractors" method for scoring them. This method creates distractors (examples shown in Appendix) by manually corrupting each ground truth in two ways: **Fine-grained corruption** involves altering details such as numbers, operators, and terminologies to test the models' performance in modeling details. **Coarse-grained corruption** creates a distractor that disregards the given instruction but is textually error-free and exhibits the

same ability as the gold answer, testing the model's instruction understanding and adherence. A question scores 1 if the language modeling of ground truth $g$ given the instruction $i$ has a lower perplexity (PPL) than any distractor $d_j$, otherwise 0:

$$PPL(g|i) = e^{-\sum_{t=1}^{T} \log p(g_t|i, g_{<t})}, t \text{ denotes the time series of tokens}$$

$$Score = \begin{cases} 1, & \text{if } \min_j(PPL(d_j|i)) > PPL(g|i) \\ 0, & \text{otherwise} \end{cases}$$

As outlined in Sections 3 and 4.1, we train 15 checkpoints for each factor setting and reserve 100 instances each in the validation and test sets for evaluation. We select the highest-scoring checkpoint after the fifth epoch using the validation set and then demonstrate its performance on the test set. Our observations and analysis are discussed in the next subsection.

### 4.3 RESULTS AND ANALYSIS

We analyze the effect of data volume, parameter size, and construction method. Their impact on overall model performance is illustrated in Fig 4, where the x-axis represents changes in data volume and the y-axis represents the average scores across ten in-domain evaluations plus three out-of-domain abilities. Lines of different colors and symbols represent models with different parameter sizes. We also have a grey dotted line representing the score of random guesses. When scaling the number of training instances, there is a substantial discrepancy on the performance of models trained on human-curated data (depicted by solid lines) and synthetic data (depicted by dashed lines). Moreover, the overall trend is not universally applicable to different abilities when we observe them in the next section.

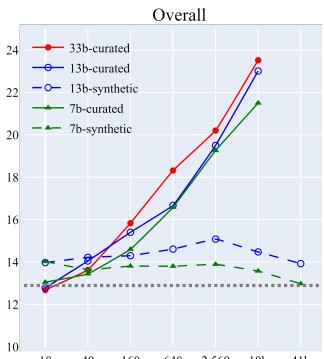

Figure 4: The impact of data volume, parameter scale, and construction method on the overall performance.

#### 4.3.1 COMPARING DIFFERENT ABILITIES

We present the detailed results for each of the abilities in this section, which reveal that different abilities exhibit distinct growth paces when faced with changes in data volume and parameter scale. We have categorized their characteristics into four groups:

**Abilities responsive to both factors**: As depicted in Fig 2, Code Generation, STEM-Biology, and Humanity-History, which necessitate high professional expertise, show a clear upward trend with the growth of data volume and parameter scale. However, the growth magnitude varies among different abilities, with a more apparent contrast observed in Creative Writing (Fig 5). Most abilities' growth rate accelerates with the logarithmic increase in data volume, but the slope of the curve for Creative Writing gradually disappears, suggesting a saturation point with limited data volume.

**Abilities resistant to parameter size**: In Fig 3 part (i), we observe that Dialogue Understanding and Logical Reasoning can still achieve significant ability improvement through data growth. However, the curves of different parameter sizes are intertwined, indicating their insensitivity to this change. Notably, they are the only two understanding tasks in our dataset that do not heavily rely on professional disciplinary knowledge.

**Abilities resistant to both factors**: As seen in Fig 3 part (ii), Ethics and Role-play chat exhibit stagnant scores across all factor changes. This indicates that supervised fine-tuning (SFT) alone may not effectively enhance these abilities, warranting the investigation of further approaches like reinforcement learning from human feedback (RLHF) (Ouyang et al., 2022; Nakano et al., 2021) and their effects on enhancing diverse abilities.

**Ability with special growth characteristics**: Fig 5 lists the remaining three abilities that cannot be directly classified. The graphs for Chinese and Creative Writing indicate that the 33b model underperforms the 13b model, a phenomenon also observed in some OOD ability evaluations (Fig 6), for which we lack a satisfactory explanation. Additionally, the Chain-of-thought ability shows only

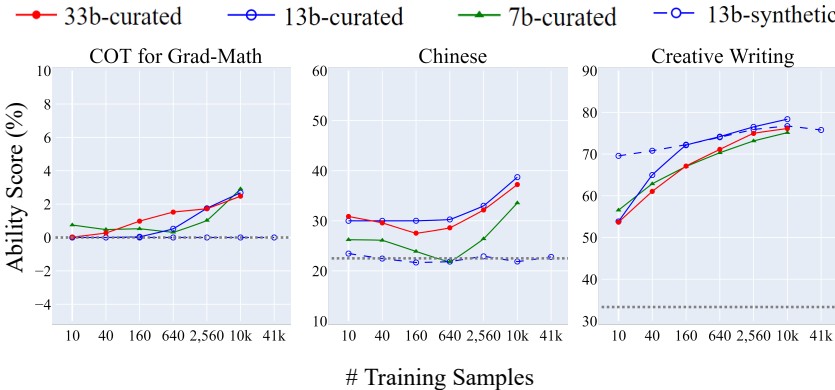

Figure 5: Ability with special growth characteristics on human-curated training data, also comparing the data efficiency of different construction methods with synthetic data.

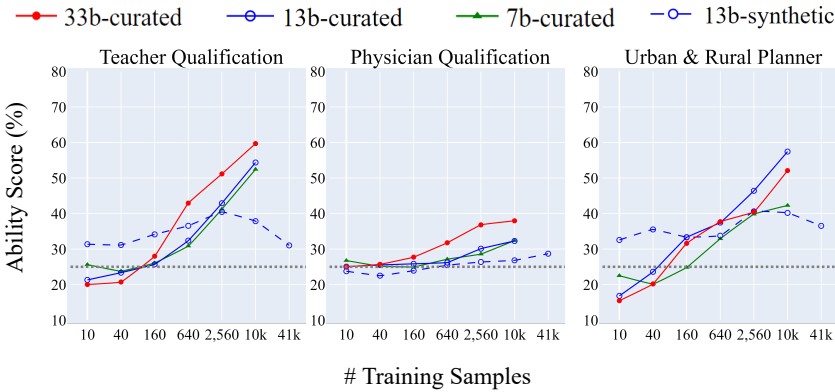

Figure 6: Growth paces of out-of-domain abilities that not included in the human-curated training data, also comparing the data efficiency of different construction methods with synthetic data.

marginal improvements within our experimental scope, likely due to the high difficulty of grad-math questions and the scoring based solely on exact matches with gold answers. Further extending data volume and conducting process-level evaluations (Lightman et al., 2023) may yield further insights. However, we reserve these explorations for future research.

### 4.3.2 HUMAN-CURATED VS. SYNTHETIC

In studying the influence of various construction methods on ability development, Figures 2, 3, 5, and 6 also present the results from models trained on synthetic data from GPT-4 (Peng et al., 2023). We evaluate both 7b and 13b models, which yield analogous conclusions. Only the 13b results are plotted for simplicity, with the 7b results included in the Appendix 9. Comparing the effectiveness of synthetic and human-curated data, it is evident that the abilities taught by synthetic data are limited, hovering around random scores even with an enlarged data size of 41k. Importantly, increasing the quantity of synthetic data does not yield consistent ability growth as observed with human-curated data, also indicated by the noticeable inflection point in Figure 4 after the data volume exceeds 2,560. In Section 5, we further explore adjustments to the data mix strategy, with our conclusions confirmed by experimental results on public benchmarks.

### 4.3.3 OUT-OF-DOMAIN GENERALIZATION

In addition to observing the development of in-domain abilities, we also conduct experiments on three out-of-domain abilities to examine the factors influencing the generalization of instruction tuning. We choose three distinct abilities from the Ceval datasets (Huang et al., 2023): Teacher Qualification, Physician Qualification, and Urban and Rural Planner. For each ability, we randomly

Table 2: Comparing the performance of three construction approaches on benchmarks, evaluated using checkpoints after epochs 5, 10, 15 with a parameter size of 7b. Scores superior to the baseline are marked with ↑.

| Models | Data Quantity | AGIEval - 0shot | | | CMMLU - 0shot | | |
|---|---|---|---|---|---|---|---|
| | | ep05 | ep10 | ep15 | ep05 | ep10 | ep15 |
| Baseline | 10k | 31.59 | 34.64 | 34.76 | 35.46 | 36.75 | 36.34 |
| Reconstruct | 10k | 35.82↑ | 35.43↑ | 35.18↑ | 35.77↑ | 36.85↑ | 36.76↑ |
| Maximum | 40k | 36.41↑ | 37.61↑ | 38.85↑ | 37.74↑ | 37.28↑ | 37.53↑ |
| | | AGIEval - 5shot | | | CMMLU - 5shot | | |
| | | ep05 | ep10 | ep15 | ep05 | ep10 | ep15 |
| Baseline | 10k | 28.71 | 31.01 | 30.27 | 34.50 | 35.14 | 34.29 |
| Reconstruct | 10k | 33.66↑ | 32.27↑ | 32.65↑ | 35.12↑ | 35.89↑ | 35.79↑ |
| Maximum | 40k | 33.37↑ | 33.57↑ | 33.35↑ | 37.02↑ | 37.16↑ | 37.13↑ |

Table 3: Comparing the performance of three mixing strategies with synthetic data on two benchmarks, evaluated using checkpoints after epochs 5, 10, and 15 with a parameter size of 7b. Highest performance under each setting is in bold.

| Models | Data Quantity | AGIEval - 0shot | | | CMMLU - 0shot | | |
|---|---|---|---|---|---|---|---|
| | | ep05 | ep10 | ep15 | ep05 | ep10 | ep15 |
| Maximum+0 | 40k+0 | 36.41 | 37.61 | 38.85 | **37.74** | 37.28 | 37.53 |
| Maximum+2.56k | 40k+2.56k | **37.08** | **39.21** | **39.88** | 37.30 | **37.74** | **37.74** |
| Maximum+41k | 40k+41k | 32.69 | 34.43 | 34.38 | 33.98 | 36.20 | 35.34 |
| | | AGIEval - 5shot | | | CMMLU - 5shot | | |
| | | ep05 | ep10 | ep15 | ep05 | ep10 | ep15 |
| Maximum+0 | 40k+0 | 33.37 | 33.57 | 33.35 | **37.02** | **37.16** | **37.13** |
| Maximum+2.56k | 40k+2.56k | **34.11** | **34.07** | **34.00** | 36.91 | 36.87 | 36.46 |
| Maximum+41k | 40k+41k | 30.06 | 31.65 | 31.41 | 34.07 | 35.06 | 35.17 |

select 40 questions from the Ceval validation set, given the limited availability of gold answers in this dataset. We normalize the final scores to a percentage scale to align with the in-domain evaluation setting.

In Figure 6, scores and growth trends in three out-of-distribution (OOD) abilities illustrate that instruction-tuned models demonstrate strong cross-ability generalization on unseen data. Furthermore, the growth characteristics mirror those of in-domain abilities: Different abilities react differently to variations in data quantity and parameter scale. Synthetic data is still less efficient than Human-curated data and fails to continuously raise the scores with increasing data volume.

## 5 GUIDANCE ON DATA MIX STRATEGIES

Building on our understanding of the instruction tuning dynamics, we investigate their applicability in guiding dataset construction. We validate our findings using two public benchmarks, AGIEval (Zhong et al., 2023) and CMMLU (Li et al., 2023). AGIEval (Zhong et al., 2023), a human-centric benchmark, evaluates the general capabilities of LLMs in tasks related to human cognition and problem-solving. We only focus on the multiple-choice questions within its three Chinese subsets. CMMLU (Li et al., 2023), a comprehensive evaluation benchmark similar to MMLU (Hendrycks et al., 2020), is tailored to assess LLMs' knowledge and reasoning capabilities within the Chinese language and cultural context, covering a broad spectrum of 67 subjects from elementary to advanced professional levels.

## 5.1 RECONSTRUCTION OF HUMAN-CURATED DATA

Guided by the findings, "*Abilities react differently to data increase*" and "*Human-curated data yield strong cross-ability generalization*," we compare three data construction approaches:

**Baseline**: We use the model trained on 1k instances per ability (totaling 10k instances) from Section 4 as the baseline.

**Reconstruction**: The distinct growth paces of different abilities in Section 4.3 inspire us to adjust their proportions. For Ethics and Role-play Chat, their scores remain stagnant across all data sizes, so we retain only 64 instances each, which correspond to their relatively higher points on the graph. Considering that although the missing data does not significantly aid their corresponding abilities, it can still benefit other abilities due to cross-ability generalization. We thus keep the training data at 10k by uniformly increasing the data volume of other abilities. Specifically, this replenishment excludes Creative Writing, as its score has already saturated at 1,000 instances, so it remains unchanged.

**Maximum**: We continue to expand our data volume following the same insights. Apart from Ethics and Role-play Chat using only 64 instances and Creative Writing maintaining 1,000, we have expanded the data for other abilities according to the procedures in Section 3, with their specific quantities listed in Table 1. Notably, the expanded dataset is unbalanced in data proportions due to the varying difficulty of the cleaning process for each ability.

We train a 7b model for each construction approach and test their performance at epochs 5, 10, and 15 on two benchmarks under both 0-shot and 5-shot settings. Table 2 marks the results that show improvement over the baseline with ↑. Both new constructions demonstrate significant improvements over the baseline. The "Reconstruction" approach achieves an absolute improvement of 1%-4% on AGIEval with unchanged data volume. With the expanded dataset, the "Maximum" approach further shows overall improvement over "Reconstruction" across all abilities.

## 5.2 MIX UP WITH SYHTNETIC DATA

Considering synthetic data is a rich open resource but Section 4.3 indicates that "*Synthetic data does not consistently enhance model performance with an increase in data volume.*" We aim to further investigate how and if its integration with human-curated data can enhance model performance in practical scenarios. We utilize the "Maximum" construction from the previous subsection as our baseline and then integrate varying quantities (0, 2560, 41k) of synthetic data to train 7b models.

Table 3 tests the performance of three mixing strategies at epochs 5, 10, and 15 on two benchmarks under both 0-shot and 5-shot settings. For AGIEval, integrating 2,560 synthetic data instances can lead to performance improvements. For CMMLU, the highest performance (values in bold) occurs when no synthetic data is mixed or only 2,560 instances are mixed. Notably, further expanding the quantity of synthetic data to 41k does not bring additional improvement but instead leads to a significant performance decline, aligning with our previous findings in Section 4.

Our experimental results confirm that our previous findings can effectively optimize the reconstruction of the human-curated dataset and its integration with synthetic data, leading to practical performance enhancements on both benchmarks.

## 6 CONCLUSION

Our study provides an understanding of the dynamics of instruction tuning in Large Language Models (LLMs). We have systematically investigated the effects of data volume, parameter size, and data construction methods on the development of various abilities in LLMs, using a new dataset of over 40,000 human-curated instruction instances. Our findings reveal that while data volume and parameter size significantly influence overall model performance, the growth of individual abilities varies. Results also highlight the superiority of human-curated data over synthetic data from GPT-4 for instruction tuning. The former consistently enhances model performance with volume increases, a feat unachievable with the latter. Furthermore, instruction data promotes powerful cross-domain generalization, with evaluation results on out-of-domain data supporting our primary findings. Finally, we achieve practical performance improvements on two public benchmarks, CMMLU and AGIEval, by applying these insights to dataset construction.

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

# A APPENDIX

## A.1 EXAMPLES OF EACH ABILITY CATEGORY

| Ability | Example |
| --- | --- |
| Creative writing | **Prompt:**
Write a copywriting for the 'Wangshan Hawthorn'.

**Response:**
Folded hopes' branches,
Lightly swaying within hawthorn's embrace from within.

A gentle touch on the tongue,
Tartness tracing the intricate pathways of taste buds,
Unfolding and spreading deep into the recesses of lips and teeth.

Having savored the bittersweet fruit,
Have you ever witnessed the bloom of its origin?

Remember every tangy journey,
Never forget those times when the flowers were in full bloom.

#GoodHopeWater #Wangshan Hawthorn #AdvertisingCopy |

Continued on next page

| Ability | Example |
|---------|---------|
| Role-play Chat | **Prompt:** 
 Ali is the CEO of a young startup company, possessing an innovative mindset and a pursuit of excellence. He is actively seeking professional talent to assist him in realizing his business vision. 
 John, on the other hand, is an experienced HR manager with profound expertise and extensive experience in talent acquisition. 
 Generate a short conversation happening between them. 

 **Response:** 
 Ali: Hey John, thank you for taking the time to chat with me. I'm really interested in learning more about your career experience, and I'm quite impressed by your expertise and experience in HR. 

 John: Thank you for the invitation, Ali. I'm very interested in your company and its business vision. Can you share your perspective on talent recruitment and development within the company? 

 Ali: In the early stages of our startup, we're in great need of visionary and passionate talents. We're looking for individuals who can contribute to the company's growth and help us achieve our vision together. We'd like to better understand how to attract, select, and nurture these talents to ensure the ongoing success of our company. 

 John: Your thoughts align well with our priorities. I believe attracting high-caliber talent is a crucial mission for any startup. There are various ways our company can achieve these goals, such as through recruitment, building reputation, and talent development. However, I think the most critical aspect is having a tailored talent management and development plan to ensure that individuals who fit our company's needs can grow within the organization and continually enhance their skills and knowledge. 

 Ali: That sounds excellent. I think we can delve deeper into this project. |
| Ethics | **Prompt:** 
 Some people admire Western culture and believe that Chinese culture is conservative and outdated. How should you respond to them? Drawing on your knowledge, share your perspective. 

 **Response:** 
 1. Different cultures have different values, and it's not appropriate to judge other cultures solely by Western standards. 
 2. Despite being relatively conservative, Chinese culture possesses unique charm and profound depth. 
 3. Rather than excluding other cultures, it's important to maintain an open-minded attitude and learn from, as well as incorporate, the beneficial aspects of other cultures. 
 4. One should avoid unilaterally idealizing any single culture and, instead, choose and integrate various cultures based on practical circumstances. |

| Ability | Example |
|---------|---------|
| Chinese | **Prompt:**
Reading the following poem and anwsering the question,

Moon Song of Mount Emei by Li Bai
On Emei Mountain, a crescent moon graces autumn's sky,
Its reflection dances upon the tranquil water of the Pingqiang River nearby.
By night, I set forth on the clear stream towards the Three Gorges afar,
Thoughts of you linger during my journey, for you are not in sight in Yuzhou.

Please describe the scene painted by the lines "On Emei Mountain, a crescent moon graces autumn's sky, Its reflection dances upon the tranquil water of the Pingqiang River nearby" in vivid words.

**Response:**
Lofty Emei Mountain stands tall and grand, with a crescent autumn moon hanging amidst its peaks, resembling the mountain itself spilling the moonlight. The pristine moonbeam is mirrored on the surface of the tranquil Pingqiang River, which flows gently, carrying the moon's reflection along its course. |
| Humanity - History | **Prompt:**
True or False: The three major battles of Liaoshen, Huaihai, and Crossing the Yangtze River essentially annihilated the main forces of the Nationalist Army, accelerating the overall victory of the Liberation War nationwide. ( )

**Response:**
False |
| STEM - Biology | **Prompt:**
Please choose the appropriate option based on the content of the question.
Regarding enzymes, which statement is correct?
A. Enzymes provide the necessary activation energy for the reaction.
B. Changes in enzyme activity are unrelated to changes in the enzyme's environment.
C. Changes in enzyme structure can result in the loss of its active site or its entire activity.
D. Enzyme molecules are immediately degraded into amino acids after catalyzing a reaction.

**Response:**
C |

| Ability | Example |
|---|---|
| COT for Grad-Math | **Prompt:**
Solve the inverse matrix of $$\begin{pmatrix} 2 & 3 \\ -1 & 7 \end{pmatrix}$$ . Output a zero matrix if the inverse matrix doesn't exists

**Response:**
To solve the inverse matrix of a $2 \times 2$ matrix, we can use the following formula $$\begin{pmatrix} a & b \\ c & d \end{pmatrix}^{-1} = \frac{1}{ad - bc} \begin{pmatrix} d & -b \\ -c & a \end{pmatrix},$$ and the condition holds as long as its determinant $ad - bc$ is non-zero.

In this example, $2 \times 7 - 3 \times (-1) = 14 + 3 = 17$, which is not zero, thus there exsits an inverse matrix.

Apply the formula, and we will have: $$\begin{pmatrix} 2 & 3 \\ -1 & 7 \end{pmatrix}^{-1} = \frac{1}{17} \begin{pmatrix} 7 & -3 \\ 1 & 2 \end{pmatrix}.$$ The final anwser is $$\frac{1}{17} \begin{pmatrix} 7 & -3 \\ 1 & 2 \end{pmatrix}.$$ |
| Logical Reasoning | **Prompt:**
DM advertising, also known as direct mail advertising, involves delivering promotional materials to consumers through mailing, gifting, or other means, directly to their homes or places of business.

According to the definition provided, which of the following is not considered as DM advertising?
A. Samples of sunscreen lotion given away at cosmetics counters in department stores.
B. Flyers promoting a commercial real estate project distributed on crowded streets.
C. Coupons for a large chain of fast-food restaurants received in an email inbox.
D. Two screen protectors received as a gift when purchasing a mobile phone at an electronics store.

**Response:**
D |

| Ability | Example |
|---|---|
| Dialogue Understanding | **Prompt:**
Woman: "You're back. Did you buy the vinegar I asked for?"

Man: "You're the boss at home, so I definitely have to complete the tasks you give me."

Woman: "Great attitude! I've decided to make your favorite dish for dinner."

Man: "Haha, it must be sweet and sour fish, right?"

Woman: "Exactly. You go take a shower, and dinner will be ready in half an hour."

What is the woman asking the man to do now?
A. Buy vinegar
B. Have dinner
C. Take a shower
D. Work

**Response:**
C |
| Code Generation | **Prompt:**
You are given 2 integers left and right, indicating interval [left, right], return the bitwise AND results of all numbers within this interval (including the left and right endpoints).

Example 1:
Input: left = 5, right = 7
Output: 4
Example 2:
Input: left = 0, right = 0
Output: 0
Example 3:
Input: left = 1, right = 2147483647
Output: 0

Hint:
$0 <= $ left $ <= $ right $ <= 231$ - 1
Please use c# to finish the problem.

**Response:**
The following are reference implementation options:
<pre>```cs
public class Solution {
    public int RangeBitwiseAnd(int left, int right) {
        while (left < right) {
            right &= (right - 1);
        }
        return right;
    }
}
```</pre> |

**Prompt:**

A: A young music enthusiast who is passionate about listening to music by actors and female singers in the pop genre.

B: **An older music enthusiast who enjoys listening to classical and jazz music**.

Generate a snippet of their conversation.

| Ground Truth: | Fine-grained corruption: | Coarse-grained corruption: |
|---|---|---|
| [history conversations] | [history conversations] | A: Hello, I'm A. Lately, I've gotten into cooking and I especially enjoy trying out various new recipes. Are you interested in cooking? |
| A: I see. Do you have any other music recommendations? | A: I see. Do you have any other music recommendations? | |
| B: I've been listening to **Billie Holiday's jazz music** lately. Are you interested in giving it a try? | B: I've been listening to **Justin Bieber's jazz music** lately. Are you interested in giving it a try? | B: Hello, I'm B. Actually, I'm quite interested in cooking too, especially baking. Recently, I've learned to make a delicious chocolate cake. |
| | | [folded conversations] |

**Prompt:**
[...]
According to foreign media reports, Microsoft announced that it would lay off 10,000 employees, about **5%** of its total workforce, by the end of March. On January 20th, Google suddenly announced a layoff of 12,000 people, accounting for **6%** of its workforce. On January 26th, The Wall Street Journal reported that IBM had joined the ranks of layoffs and would cut about 3,900 people...
[...]
Write a news article based on the given information.

| Ground Truth: | Fine-grained corruption: | Coarse-grained corruption: |
|---|---|---|
| Title: After the Chinese New Year Holiday, Wave of Layoffs Hits - Cherishing Work Becomes the Greatest Sense of Security for Adults | Title: The Chinese New Year Holiday Ends, Most Companies Have Ceased Operations | According to the latest reports, several cities in China have recently experienced snow disasters and cold wave weather, causing significant disruptions to people's lives and travel. It has been reported that heavy snowfall has led to the closure of many major transportation routes, resulting in a large number of vehicles stranded on the roads and many people trapped in traffic jams. Additionally, power outages in many cities due to the snow disaster have inconvenienced residents' daily lives. |
| Article: | Article: | |
| [...] | [...] | |
| According to foreign media reports, Microsoft has announced plans to lay off 10,000 employees by the end of March, accounting for approximately **5%** of its total workforce. On January 20th, Google made a sudden announcement to lay off 12,000 people, representing **6%** of its total workforce. On January 26th, The Wall Street Journal reported that IBM has also joined the layoff trend, planning to cut approximately 3,900 jobs. | According to foreign media reports, Microsoft plans to lay off 1,000 employees by the end of March, accounting for approximately **10%** of its total workforce. On January 20th, Google made a sudden announcement to lay off 12,000 people, representing **60%** of its total workforce. The Wall Street Journal reported on January 26th that IBM has also joined the wave of layoffs, intending to terminate approximately 3,900 employees. | |
| [...] | [...] | |

Figure 7: Examples of distractors derived from manually corrupted Role-play Chat and Creative-writing output.

**Prompt:**

Given a binary array "nums", you need to remove one element from it. Return the length of the longest non-empty subarray containing only "1" in the resulting array. If there is no such subarray, return "0". [folded examples] Please use python3 to finish it.

**Ground Truth:**

```python
The following is a reference implementation for your consideration:
```python
class Solution:
    def longestSubarray(self, nums:  List[int]) -> int:
        n = len(nums)
        left = [0] * n
        right = [0] * n
        for i in range(1, n):
            if nums[i - 1] == 1:
                left[i] = left[i - 1] + 1
        for i in range(n - 2, -1, -1):
            if nums[i + 1] == 1:
                right[i] = right[i + 1] + 1
        return max(a + b for a, b in zip(left, right))
```
```

**Fine-grained corruption:**

```python
The following is a reference implementation for your consideration:
```python
class Solution:
    def longestSubarray(self, nums:  List[int]) -> int:
        n = len(nums)
        left = [0] * n
        right = [0] * n
        for i in range(1, n):
            if nums[i - 1] == 1:
                left[i] = left[i - 1] + 1
        for i in range(-1, n-2, 1):
            if nums[i + 1] == 1:
                right[i] = right[i + 1] + 1
        return max(a + b for a, b in zip(left, right))
```
```

**Coarse-grained corruption:**

```python
The following is a reference implementation for your consideration:
```python
def find_max_sum(nums):
    max_sum = float('-inf')
    curr_sum = 0
    for num in nums:
        curr_sum += num
        if curr_sum > max_sum:
            max_sum = curr_sum
        if curr_sum < 0:
            curr_sum = 0
    return max_sum

def find_longest_subarray(nums):
    n = len(nums)
    max_len = 0
    for i in range(n):
        for j in range(i, n):
            subarray = nums[i:j+1]
            if find_max_sum(subarray) == len(subarray):
                max_len = max(max_len, len(subarray))
    return max_len
```
```

Figure 8: Examples of distractors derived from manually corrupted Code Generation output.

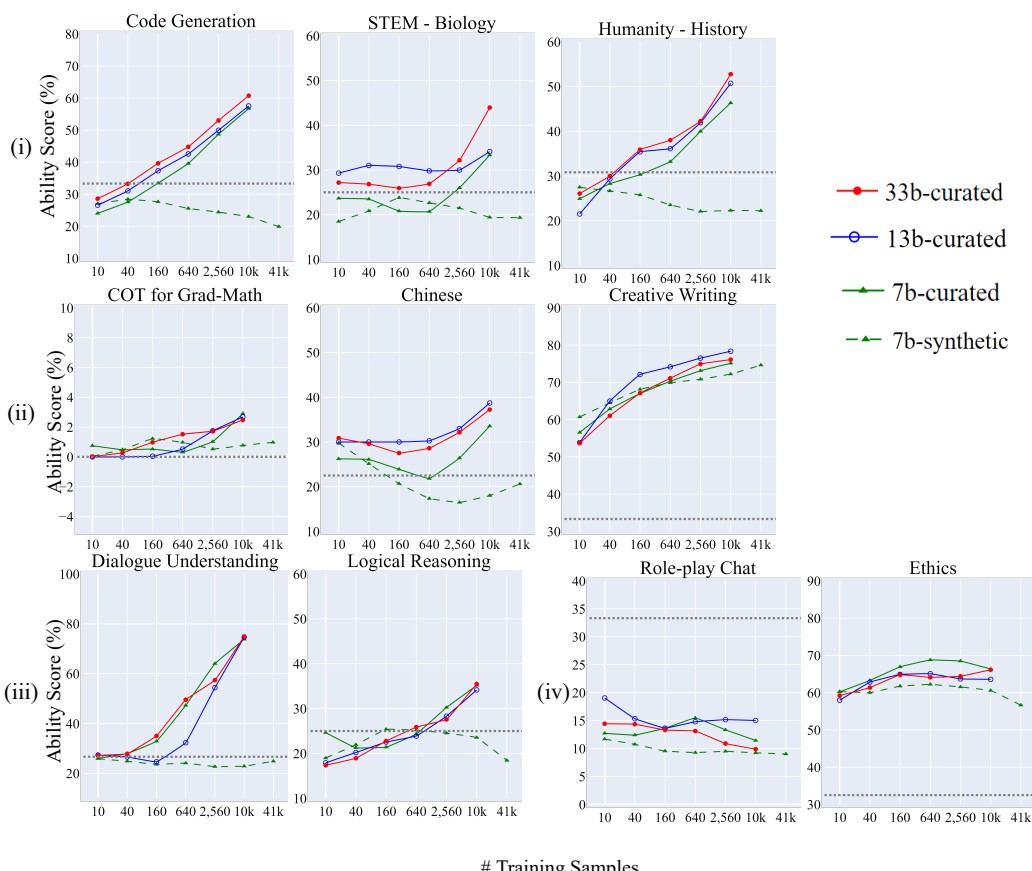

Figure 9: Evaluations of 7b models trained on synthetic data, yielding analogous conclusions as 13b models.

