# OpenReview forum: "Dynamics of Instruction Tuning: Each Ability of Large Language Models Has Its Own Growth Pace"
_ICLR.cc/2024/Conference — ICLR 2024 Conference Withdrawn Submission_

### Official Review · Reviewer_ezqx · 2023-10-31

**Soundness:** 2 fair
**Presentation:** 3 good
**Contribution:** 2 fair
**Rating:** 5
**Confidence:** 4

**Summary:**

This paper studies the scaling rate of specific capabilities in instruction fine-tuned language models as a function of model size, dataset size, and source (human vs synthetic). In order to facilitate the study, the paper introduces a "new human-curated Chinese dataset, comprising more than 40,000 instruction-output pairs, each subject to stringent quality control" which is assembled from existing sources (C3-D, PRM800K, COIG, etc) and ≈ 1K additional examples of "Creative Writing User Queries from In-House Data". Experiments with this dataset show some abilities are learned easily using limited data regardless of model size, while other abilities are relatively unresponsive to both factors. Additional experiments show synthetic data is less helpful overall.

**Strengths:**

1. Better understanding the relationship between instruction tuning data characteristics and downstream performance is an important topic. The paper presents extensive experiments to help better understand how data composition (in terms of capability density) relates to downstream performance.
2. Although the dataset here is assembled primarily from existing sources, the standardization, filtering, and revision required significant manual effort. The release of clean non-English instruction tuning data is a valuable contribution that could benefit future work.

**Weaknesses:**

1. The presentation of the experiments with synthetic data is unclear. I believe the dataset used is from [1], but it would be helpful to make this clear and for the paper to include some details about the dataset for the sake of remaining self-contained. Additionally, as far as I can tell this dataset is not balanced with respect to capability frequency in the same way the human-authored data is. This makes it unclear the extent to which these results are meaningfully comparable.
2. I find the evaluation methodology used for "Open-ended questions" questionable. While the probability-based ranking is intuitive, fine-tuned LMs are known to be poorly calibrated [2]. As a result, I am not convinced a model's ability to rank a specific set of pre-specified completions by perplexity correctly necessarily reflects performance characteristics in real-world settings. The random (or close to random) performance in many settings (all models are worse than random for "Role-play Chat") reinforces these concerns.
3. Improving non-English instruction tuned LMs based on English foundation models is an important direction. Unfortunately, it is also a potential confounder for the type of broad conclusions drawn in this paper. It seems plausible that many of the conclusions could be explained due to this discrepancy. This issue should be addressed in the paper.
4. The out-of-distribution capabilities don't seem like the same kind of "capabilities" as the in-distribution capabilities, e.g., "Creative Writing" and "Logical Reasoning" are very different in breadth/scope than "Urban & Rural Planning".
5. The paper makes broad claims regarding the relative value of synthetic data that are not well supported. Numerous past works (cited in this paper) show clear improvements using synthetic data. I would expect such claims to be supported by experiments on more datasets and consider more than a single synthetic method.

**Other Issues**

1. I'm confused by the sentence "We normalize the final scores to a percentage scale to align with the in-domain evaluation setting." in section 4.3.3. I assumed all figures were percentages since they are labeled "Ability Score (%)".
2. I find Figure 1 inappropriate for a scientific publication. I'm surprised there wasn't something more interesting to put here given page limits.

**References**

1. Baolin Peng et al. "Instruction Tuning with GPT-4." 2023.
2. Percy Liang et al. "Holistic Evaluation of Language Models." 2022.

**Questions:**

See weaknesses.

---

### Official Review · Reviewer_e9FS · 2023-11-01

**Soundness:** 2 fair
**Presentation:** 2 fair
**Contribution:** 2 fair
**Rating:** 3
**Confidence:** 4

**Summary:**

The paper uses a corpus of 10 datasets for instruction tuning mainly in Chinese to study how sensitive language models are to instruction tuning data size and model size across different tasks (or abilities). The study shows that different abilities have different sensitivities to these factors (data and model size). Another contribution is a set of human curated instruction data for the same corpus.

**Strengths:**

S1 - Studying the impact and sensitivity of instruction tuning in and out of domain is an important and timely problem.

S2 - The work makes a good effort to provide relevant model checkpoints and create new human curated data for instruction tuning.

**Weaknesses:**

W1- While the work quantifies model behavior and sensitivity to data and model size, it does not provide further insights to why some abilities are more or less sensitive to these factors. The observation that each ability has its own growth pace is somewhat not surprising and a deeper contribution would follow up with investigating the reasons behind different behaviors or categorizing them.

W2- The protocol of dataset selection for instruction tuning needs some analytical justification to why this is a good set of tasks to pick and how they are qualitatively different from each other.

W3- Out of domain improvements in Table 2 look modest (less than 4%). In many ways, one could also use this as an indicator of the generality of the instruction tuning corpus. Does this indicate that out of domain generalization still remains an open problem even for instruction tuned LLMs?

**Questions:**

Q1- Ability scores for Grad-Math appear to be pretty low. Does it still make sense to present these results?

**Details Of Ethics Concerns:**

It is necessary to describe the process followed for human annotation, IRB, ethical reviews, and fair pay for the region where the data was collected.

I'd also recommend the authors to change the dialogue conversation in page 18 as it reinforces stereotypes of women cooking and the such.

---

### Official Review · Reviewer_29zA · 2023-11-01

**Soundness:** 3 good
**Presentation:** 3 good
**Contribution:** 3 good
**Rating:** 6
**Confidence:** 3

**Summary:**

Previous studies offer mixed conclusions about the number of instructions required. To gain insights into data construction, the study shifts its focus from overall model performance to specific abilities like creative writing, code generation, and logical reasoning. It systematically investigates the impact of data volume, model size, and data construction methods using over 40,000 human-curated instruction data categorized into ten distinct LLM abilities. The findings indicate that while data volume and model size affect performance, some abilities respond better to these changes. Human-curated data consistently outperforms synthetic data and shows efficiency gains with more data, unlike synthetic data.

**Strengths:**

This paper provides comprehensive analysis to study the learning paces of different tasks when the models are trained with human-curated and synthetic data. It brings very interesting findings that the increase of synthetic data doesn't help to improve the performance of specific domains. Also, the observation that adding a small part of synthetic data leads to better performance than the full set may also inspire the future dataset construction for improving performance on domain-specific tasks. Overall, I really like the novel conclusions discussed in this paper. It will let more researchers to reflect the choice whether we need to fully rely on synthetic data or not.

**Weaknesses:**

Despite the comprehensive analysis, there are several weaknesses:

**Insufficient Fine-Grained Ablation Studies**: The absence of more detailed ablation studies is evident in the paper. Given the inclusion of multiple domains in the training data, it would be valuable to assess how the model's performance on a domain A dataset is influenced when additional domain-specific datasets, such as B and C, are introduced. These studies could shed light on whether the inclusion of such datasets hinders or expedites the learning process.

**Limited Insights into Phenomena**: The paper provides numerous findings and conclusions, but it lacks an in-depth exploration of the underlying causes behind these phenomena. While the paper presents interesting observations, readers are left wanting a deeper discussion to enhance the quality of the paper.

**Novelty Constraints**: It's worth noting that the efficiency of synthetic data has been previously discussed in Gudibande et al., 2023, in their paper titled "The False Promise of Imitating Proprietary LLMs."

**Questions:**

1. Why do you focus on Chinese datasets, instead of English? There are already lots of English instruction tuning data. It is also easy to convert some domain-specific datasets into instruction-tuning data format by manually annotating the instructions. Is there any difference between English and Chinese task evaluation?

2. One missing reference that also discusses the dynamics of instruction tuning - Dynosaur: A Dynamic Growth Paradigm for Instruction-Tuning Data Curation. Yin et al., 2023. This paper proposes an automatic instruction generation method that can support the dynamic expansion of the instruction-tuning data. Also, it discusses how we can further improve the performance on unseen tasks and avoid forgetting on existing trained tasks, when the instruction data dynamically grows. Moreover, it can be counted as a combination of "human-curated" and "synthetic" data, since it is based on human annotations but instructions are generated with machines.

---

### Official Review · Reviewer_G2Rz · 2023-11-01

**Soundness:** 2 fair
**Presentation:** 3 good
**Contribution:** 2 fair
**Rating:** 3
**Confidence:** 4

**Summary:**

This paper investigates the effects of data volume, parameter size, and data construction methods on ten aspects of LLMs abilities. They collect 10k human-curated data in total and aggregated them with synthetic data. They draw several conclusions such as real data benefits LLMs more compared to synthetic data; Different abilities of models require different amounts of training.

**Strengths:**

1. Some observations are useful. For example, scaling up data size would clearly benefit Code Generation, STEM-Biology.
2. The experimental design is comprehensive. The paper investigates different size of the dataset size, split the categories in details, compare different model size and checkpoints.

**Weaknesses:**

1. Although the paper investigates many facets of influencers to instruction tuning models, the conclusions look trivial to me. Most conclusions are simply empirical observations. I expect to see more in-depth analysis. For example, why
2. More baselines needed. The paper work on Chinese datasets of Instruction Tuning. As an analysis paper, we should try to make sure the comparisons are made under the SOTA performance, otherwise, it would be less helpful. I'm not quite familiar with the baseline performance on those Chinese datasets. This will affect the quality of the analysis.
3. I'm concerned about the amount of data for each ability (1k), which looks small to me. Usually, the English IT benchmarks have around 60K data points. The abilities being investigated are not some superficial abilities of the models, mostly involve reasoning. Therefore, I doubt whether 1k is enough for each ability. It can also be seen from Figure 4 that the performance increase has not converged (the performance has not reached a plateau), which means more data would be helpful and might change some claims they make. Some claims under this setup thus, might not be truthful.
4 The parts in the paper about mixing up seem to be rushed.
5. The plots can be confusing. The authors draw human-curated + synthetic plots in one figure. This sometimes confuses people at first glance.

**Questions:**

1. Maybe I missed it, but I cannot find the details about how to collect the synthetic data. Can you elaborate on it? Without the details, it will be hard to understand why synthetic settings are so much worse than real data, which also contradicts the observations in English datasets (self-instruct, Dynasour ...).
2. Why we only use 10k human-curated data as from Figure 4 the performance has not yet converged.